# Liquid Biopsy for Oral Cancer Diagnosis: Recent Advances and Challenges

**DOI:** 10.3390/jpm13020303

**Published:** 2023-02-08

**Authors:** Yutaka Naito, Kazufumi Honda

**Affiliations:** 1Department of Bioregulation, Institute for Advanced Medical Science, Nippon Medical School, Tokyo 113-8602, Japan; 2Department of Bioregulation, Graduate School of Medicine, Nippon Medical School, Tokyo 113-8602, Japan

**Keywords:** oral cancer, oral squamous cell carcinoma, liquid biopsy, circulating tumour cells, microRNAs, extracellular vesicles, cell-free DNAs

## Abstract

“Liquid biopsy” is an efficient diagnostic tool used to analyse biomaterials in human body fluids, such as blood, saliva, breast milk, and urine. Various biomaterials derived from a tumour and its microenvironment are released into such body fluids and contain important information for cancer diagnosis. Biomaterial detection can provide “real-time” information about individual tumours, is non-invasive, and is more repeatable than conventional histological analysis. Therefore, over the past two decades, liquid biopsy has been considered an attractive diagnostic tool for malignant tumours. Although biomarkers for oral cancer have not yet been adopted in clinical practice, many molecular candidates have been investigated for liquid biopsies in oral cancer diagnosis, such as the proteome, metabolome, microRNAome, extracellular vesicles, cell-free DNAs, and circulating tumour cells. This review will present recent advances and challenges in liquid biopsy for oral cancer diagnosis.

## 1. Introduction

Recent technological advances in multi-omics platforms have improved our understanding of tumour heterogeneity [1,2,3,4,5]. The genetic background of cancer cells differs significantly among patients. The physiological state of tumours changes over time and is influenced by various factors, including metabolic and homeostatic mechanisms, surrounding microenvironment factors, and drug selection pressures [6]. Such intra-tumoral and interpatient heterogeneity contributes to treatment failure in patients with cancer [7,8]. Therefore, capturing accurate tumour information in each individual is essential for early detection, determining treatment protocols, predicting disease progression, and predicting treatment efficacy.

“Liquid biopsy” is a test for analysing biomaterials (proteins, microRNAs (miRNAs), circulating tumour cells (CTCs), etc.) in body fluids, such as blood, saliva, breast milk, and urine (Figure 1) [9]. Cancer cells and stromal cells within the tumour tissues can secrete various molecules in body fluids [9,10]. Cancer cells detach from tumour tissue during the process of metastasis and circulate in the bloodstream [11]. A biomarker is a factor that is an indicator for normal biological processes, pathogenic processes, or pharmacologic responses to a therapeutic intervention [12]. Biomaterials in body fluids can be considered biomarkers if they reflect such physiological and pathological states of tumours. Liquid biopsy detects these biomarkers, which can include proteins, metabolites, glycans, nucleic acids, and cells in body fluids, and provides “real-time” information about individual tumours (Figure 1). Therefore, liquid biopsy has been studied widely over the past two decades as an attractive diagnostic tool [9,13]. Liquid biopsy is non-invasive and more easily repeatable than conventional histological analysis. So far, many molecular candidates have been investigated for liquid biopsy in cancer diagnosis, such as proteome, microRNAome (miRNAome), glycome, extracellular vesicles (EVs), cell-free DNAs (cfDNAs), and circulating tumour cells (CTCs) [9,10,14].

Head and neck cancer is the seventh most commonly diagnosed solid malignancy worldwide [15,16]. Within head and neck cancer, oral cancer (OC) arises from the oral cavity, including the buccal mucosa, anterior tongue, floor of the mouth, hard palate, upper and lower gingiva, and retromolar trigone. Histologically, approximately 90% of oral cancer is squamous cell carcinoma (SCC) from the mucosal epithelium [17]. Despite improvements in therapeutic strategies, the 5-year survival rate of patients with oral squamous cell carcinoma (OSCC) in later stages (Stage III and IV) remains poor [18,19]. A quick and non-invasive approach, such as liquid biopsy, would be useful for predicting oral cancer metastasis and monitoring treatment efficacy. How could liquid biopsy assist in the diagnosis of oral cancer? This review will focus on recent advances in cancer biomarker research and discuss novel biomarker candidates for liquid biopsy in oral cancer.

## 2. Liquid Biopsy for Cancer Detection

Although many potential candidates have been evaluated for the detection, prediction, and therapeutic monitoring of cancer, few have been used in clinical practice. This review will first present representative molecules being used in clinical practice before focusing on molecular candidates for oral cancer diagnosis (Table 1). In this review, we will additionally focus on the latest findings published within the last two years and those with biological evidence.

### 2.1. Conventional Cancer Markers

The levels of some molecules in blood have been tested clinically for use in cancer detection. Such molecules are termed “cancer markers”. For example, the levels of carbohydrate antigen (CA) 19-9 and CA125 [31] in patient blood can reflect abnormal glycosylation of tumour cells. Carcinoembryonic antigen (CEA) [32] and alpha-fetoprotein (AFP) [33] are well-studied oncofoetal antigens that are expressed at high levels in cancer cells and foetal tissues. Regarding specific antigens for organs or tissues, increased expressions of prostate-specific antigen (PSA) [34] and squamous cell carcinoma antigen (SCC-Ag) [35] are observed in patients compared with healthy individuals. However, the specificity and sensitivity of these cancer markers for early cancer detection remain inadequate. For instance, positivity for CA19-9 (a biomarker for gastrointestinal cancers) occurs in less than 52% of cases of pancreatic cancer under 2 cm in size. These markers are also detected in patients with pancreatitis, endometriosis, and diabetes mellitus. In addition, the carbohydrate antigenic epitope of the CA19-9 antibody is the sialyl Lewis A antigen. Some patients who lack α1-4 fucosyl transferase cannot produce the sialyl Lewis A antigen, resulting in little or no CA19-9 in their serum even with tumour development [36]. However, testing for CA19-9 has benefits for monitoring of treatment response to anticancer drugs and postoperative recurrence [37].

The usefulness of cancer biomarkers, such as CA19-9 [29], CEA [29,38], SCC-Ag [29,39], immunosuppressive acidic protein (IAP) [30], and cytokeratin 19 fragment (Cyfra), for diagnosis of oral cancer has been reported [29,40]. However, since its accuracy in diagnosis ranges from 64% to 71% [29], these markers are not applicable to all patients with oral cancer.

### 2.2. miRNAs for Cancer Detection

MicroRNAs (miRNAs), small noncoding RNAs of ~25 nucleotides in length, can be packaged into EVs [41,42] or form a complex with some proteins, such as Ago2 [43,44] and high-density lipoproteins [45], and circulate in various body fluids. Such circulating miRNAs are easily detected in a small amount of body fluid by conventional PCR or high-throughput platforms, such as microarray and next-generation sequencing. Therefore, several studies have focused on miRNAs as possible candidates for liquid biopsy for early cancer detection. In this section, we will focus on studies that have assessed the total expression levels of each target miRNA in patients’ blood and saliva samples.

Regarding the development of liquid biopsy that targets circulating miRNAs, a nationwide project in Japan (termed Development and Diagnostic Technology for Detection of miRNA in Body Fluid) was conducted to identify novel serum miRNA profiles for screening for 13 tumour types [46], including breast cancer [47,48], bladder cancer [49], ovarian cancer [50], sarcoma [51], prostate cancer [52], and hepatocellular carcinoma [53]. The project also developed a machine learning model for miRNAome-based cancer type prediction [46] that achieved high diagnostic performance, with an accuracy of 0.88 for all cancer stages and accuracy of 0.90 for earlier stages (Stage 0–II). The model could identify 13 cancer types with high accuracy. Regarding the early detection of oral cancer, Nakamura et al. found that serum miRNA expression signatures, including miR-19a, miR-20a, and miR-5100, could distinguish between OSCC patients and healthy donors with high diagnostic accuracy [20]. Romani et al. conducted a microarray analysis of salivary miRNAs and identified 25 miRNAs expressed differently between OSCC patients and healthy controls [21]. They also showed that the combination of miR-106b-5p, miR-423-5p, and miR-193b-3p predicted OSCC with an accuracy of 0.98. Mehterov et al. demonstrated that an miRNA panel consisting of miR-21, miR-93, miR-133b, miR-146b, miR-155, and miR-182 could detect OSCC with a sensitivity and specificity of 98% and 60%, respectively [54]. It is a limitation of these studies that they were case-control studies, thus the timing at which these candidate miRNAs become detectable in patients’ body fluids is yet to be determined. Further investigation using prospective cohorts will provide practical strategies for cancer detection by miRNA profiles.

### 2.3. Extracellular Vesicles for Cancer Detection

EVs are lipid bilayer membrane vesicles and include various bioactive molecules, including proteins, mRNAs, metabolites, and miRNAs. They were initially considered “garbage bins” for exporting unnecessary intracellular molecules. However, the fact that circulating miRNAs could be packaged into EVs [41,42] accelerates research on EVs as attractive targets for improving cancer treatment and diagnosis.

Several studies have focused on molecules in EVs as possible candidates for liquid biopsy for oral cancer detection. Bigagli et al. showed that miR-210 in serum-derived EVs is a potential target for early detection of OSCC with a specificity of 86.67% and sensitivity of 92.31% [22]; miR-24-3p in saliva-derived EVs could be used as a cancer biomarker, as reported by He et al. [23]; and miR-24-3p expression showed high accuracy with an area under the curve (AUC) value of 0.738. Gai et al. also identified miR-512-3p and miR-412-3p, which are up-regulated in saliva-derived EVs in OSCC patients compared to healthy individuals. Expression of both of these miRNAs in patient saliva-derived EVs predicted OSCC, with an accuracy of 0.847 and 0.871, respectively [24]. Several studies have focused on other biomolecules packaged into EVs as targets for liquid biopsy. Nakamichi et al. investigated the clinical utility of Alix included in serum-derived and saliva-derived EVs in oral cancer detection [25]. Alix is an EV marker that plays a crucial function in exosome biogenesis [55,56,57,58]. The level of Alix is higher in EVs derived from patient serum and saliva than in those from healthy controls. Alix in patient serum-derived and saliva-derived EVs shows moderate performance, with AUCs of 0.685 and 0.712, respectively [25]. The utility of proteins in EVs as prognostic biomarkers will be discussed in Section 4.1. Although these studies have been implemented in small cohorts, their findings provide a potential new avenue for the clinical application of EVs in body fluids for early cancer detection.

### 2.4. Circulating Cell-Free DNAs for Cancer Detection

Circulating cell-free DNAs (cfDNAs) are degraded double-strand DNA fragments and are detectable in human body fluids [59]. In a healthy individual, most cfDNAs are released from the hematopoietic system. However, tumour cells can also release cfDNAs (termed circulating tumour DNAs (ctDNAs)) and change the composition of cfDNA profiles. Rapado-González et al. showed that the total concentration of PCR amplicons of cfDNAs was significantly higher in the saliva of patients than of healthy controls. They targeted the Arthobacter luteus (ALU) retrotransposon sequence to detect cfDNAs, and one of these cfDNA amplicons predicted OSCC with a sensitivity and specificity of 83.33% and 73.33%, respectively [27]. Sayal et al. reported the utility of cfDNAs and cell-free mitochondrial DNA in patients’ saliva in the detection of head and neck squamous cell carcinoma (HNSCC). They also used the PCR method for cfDNA detection and reported an accuracy for discriminating HNSCC and non-HNSCC of 77.37% and 80.5%, respectively [28].

## 3. Biomarkers for Predicting Therapeutic Efficacy

Malignant tumours from the same organ can harbour different driver gene mutations between individuals, leading to differential consequences for the therapeutic response [60]. Testing for the gene mutation of cancer cells and immunohistochemistry analysis can be performed using biopsy and resected tumour specimens to predict therapeutic efficacy. Liquid biopsy is less invasive than these methods and has potential as a powerful tool for guiding individualized treatment strategies.

### 3.1. Mutation Status of Genes in Cancer Cells for Predicting Therapeutic Efficacy

Understanding the “mutation status” of genes in cancer cells enables the identification of patients who would further benefit from molecular targeting agents. For instance, epidermal growth factor receptor (EGFR)-tyrosine kinase inhibitors (EGFR-TKIs) are well-studied in the treatment of nonsmall cell lung cancers (NSCLCs). Gefitinib is a first generation EGFR-TKI that was approved by the Japanese Ministry of Health, Labour and Welfare (MHLW) in 2002 as a monotherapy treatment for inoperable or recurrent NSCLC. However, phase III trials (INTACT 1 (Iressa NSCLC Trials Assessing Combination Treatment) and INTACT 2) [61,62] failed to provide significant improvements in the outcomes of patients with NSCLC. As the leading causes of these failures, three independent studies conducted in 2004 demonstrated that EGFR gene mutations (a base-pair deletion at exon 19 and L858R point mutation at exon 21) [63,64,65] are related to dramatic responses to gefitinib. These gene mutations are frequently found in patients who are female, Asian, nonsmokers, and who have adenocarcinoma [63]. Based on these reports, a phase III clinical trial (IPASS (the Iressa Pan-Asia Study)) was conducted to compare progression-free survival (PFS) between doublet chemotherapy (carboplatin + paclitaxel) and gefitinib in patients with NSCLC. PFS was significantly longer in patients receiving gefitinib than in those receiving chemotherapy in the EGFR mutation-positive subgroup. Conversely, PFS was shorter in the EGFR mutation-negative subgroup [66]. As another EGFR gene mutation, T790M point mutation in exon 20 is also associated with resistance to EGFR-TKI. This mutation increases the affinity of EGFR for ATP, thus reducing the binding of EGFR-TKIs to the ATP pocket [67,68]. Osimertinib, a third-generation EGFR-TKI, is effective in patients with the EGFR T790M mutation. The efficacy of osimertinib in locally advanced and metastatic NSCLC patients with EGFR mutations was demonstrated in a phase III clinical trial (the FLAURA study). Osimertinib is now recommended as a first-line treatment for patients with EGFR mutations [69]. These representative studies demonstrate that oncogenic driver gene mutation status is a valuable biomarker for distinguishing between responders and nonresponders to EGFR-TKIs.

Protein expression in cancer cells can also provide essential clues to guide treatment strategies. Human epidermal growth factor receptor-2 (HER2), estrogen receptor (ER), and progesterone receptor (PgR) are used to stratify breast cancer patients who respond to anti-HER2 antibody (trastuzumab) or hormone therapy. Combined immunohistochemistry analysis for HER2 expression and fluorescence in situ hybridization for HER2 gene amplification has been implemented. HER2 overexpression is also found in salivary gland carcinoma [70]. A recent study demonstrated that a combination of trastuzumab and docetaxel improves survival in patients with salivary gland carcinoma [71]. Furthermore, the Japanese MHLW approved trastuzumab for advanced salivary gland tumours in 2021. Although patient stratification using HER2 expression in salivary gland carcinoma has not yet been implemented clinically, it could be used as a biomarker for predicting treatment response.

### 3.2. EVs for Predicting Therapeutic Efficacy

Cancer-derived factors are packaged into EVs and can be detectable in body fluids. As for HER2 status in salivary gland carcinoma mentioned above, the molecules in EVs can be possible candidate markers for predicting the efficacy of trastuzumab. Li et al. investigated the utility of HER2-positive EVs in predicting HER2 status in gastric cancer tissue, and HER2-positive EVs showed an AUC value of 0.746 for discriminating tissue HER2 status [72]. In breast cancer, Zhang et al. identified miR-1246 and miR-155 in cancer-derived EVs, which can predict trastuzumab resistance [73]. Although these studies did not investigate salivary gland tumours, EVs could be used as an aid for determining treatment strategies for oral cancer.

### 3.3. Circulating Tumour Cells and ctDNAs for Predicting Therapeutic Efficacy

CTCs are cancer cells that have been released into the bloodstream from a primary tumour, and they have an extremely small population compared to other cell populations in the blood. Nevertheless, CTCs have attracted interest as targets for liquid biopsy because they indicate biological and clinical aspects of the primary and metastatic niches of tumours.

We recently investigated whether CTCs can be used to guide clinical strategies by observing molecular changes in patients with advanced-stage cancers [74]. Using the microfluidics flow method, we collected CTCs from four different cancer types (head and neck, oesophageal, gastric, and colorectal) and performed targeted sequencing of CTCs and ctDNAs using next-generation sequencing (NGS). We combined analyses of alterations in the genomic profiles of CTCs and ctDNA from patients who underwent therapy with anti-EGFR antibodies and identified unique genetic alterations in both CTCs and ctDNAs of patients with metastatic colorectal cancer before and after treatment. Intriguingly, concordance between genetic mutation profiles of the CTCs and ctDNAs was not always observed, which suggests that the genetic alteration profiles differ between CTCs and ctDNAs. We, therefore, performed combined analyses of CTCs and ctDNAs to improve the detection of genomic alterations and identified missense mutations in 5 of 10 cases of head and neck cancer (50%) and in 15 of 18 cases of gastrointestinal cancer (83.3%). Our data demonstrated that CTCs and ctDNAs exhibited genetic heterogeneity and that both must be evaluated for optimal monitoring of disease progression and treatment selection in the clinical setting. Furthermore, the combination of NGS analysis of CTCs and ctDNAs enabled the identification of increased rates of genetic mutation in patients who exhibited resistance to anti-EGFR therapy. Mutations in codon 61 in KRAS and NRAS were detected more frequently in colorectal cancer patients with anti-EGFR therapy resistance than before the initiation of anti-EGFR therapy. These data show that liquid biopsy can identify genetic mutations related to treatment resistance and can be expanded by combining CTC and ctDNA analyses.

## 4. Liquid Biopsy for Prognostic Biomarkers of Oral Cancer

Several biomarkers that predict patient prognosis have also been reported. This section will focus on liquid biopsy for prognostic biomarkers of oral cancer.

### 4.1. miRNAs and EVs for Predicting Patient Outcomes

As mentioned above, miRNAs and EVs appear to be powerful tools for early cancer detection. Several studies have also reported their application as biomarkers for predicting patient outcomes. Romani et al. investigated whether miRNA expression profiles in patient saliva can predict prognosis in OSCC patients. Although theirs was only in a small OSCC cohort, they showed that miR-423-5p expression in patient saliva was an independent factor related to a poor prognosis in OSCC [21]. Regarding EVs, several research groups have provided evidence that EVs in patient saliva and serum can be utilised as prognostic biomarkers for oral cancer. Yamana et al. investigated the role of miR-503-3p in EVs derived from radiotherapy-resistant OSCC cells. Radioresistant OSCC-derived EVs transfer miR-503-3p to recipient cells and suppress radiation-induced apoptosis by targeting BAK expression. They also showed the clinical significance of miR-503-3p in the EVs of OSCC patients who underwent preoperative chemoradiotherapy. Patients with high miR-503-3p had higher rates of lymph node metastasis (*p* = 0.043) and had a poor pathological response to preoperative CRT (*p* = 0.003) [26]. Qu et al. performed proteome analysis by quantitative profiling using isobaric labelling (iTRAQ) and identified 43 EV-associated proteins related to lymph node metastasis of OSCC [75]. They also showed that expression of these candidates, including myosin-9, agrin, integrin alpha-3, actin, and serum paraoxonase/arylesterase 1, was associated with shorter overall survival in OSCC patients. Further validation of the robustness of these biomarkers conducive to liquid biopsy is warranted.

## 5. Limitations and Challenges of Novel Biomarkers for Oral Cancer

As discussed above, increased evidence has shown that liquid biopsy can be a powerful tool for oral cancer diagnosis. For other cancer diagnoses, such as pancreatic cancer and colorectal cancer, several clinical trials have been conducted (Cologuard [76], Galleri test [77], Guardant Health [78]). However, unfortunately, no biomarker is approved by the US Food and Drug Administration (FDA) for oral cancer diagnosis. Tumour lesions can be more readily assessed visually in oral cancer than in other cancer types due to easy access to the oral cavity. Therefore, a liquid biopsy might be less advantageous for detecting tumour lesions and predicting treatment efficacy than for other cancers. In the clinical setting, a screening program for oral cancer is not recommended because it is considered that the frequency of oral cancer is very rare. Incidental early detection of oral cancer can occur during dental treatment and oral healthcare. Recently, contactless cancer screening programs have become of interest in the asymptomatic population because of the expansion of novel infectious diseases. If a multicancer screening method, for not only oral cancer but also other cancers, could be carried out using liquid biopsies during a single clinical examination, it would have a positive impact on precision cancer prevention in a contactless environment. Thus, the establishment of multicancer screening methods using liquid biopsies is very important for cancer prevention. Moreover, there is potential for biomarkers to be used for the detection of late metastatic cervical lymph nodes and distant metastases that cannot be seen macroscopically. There are also technical issues, such as regarding how to increase the robustness of the analysis results and the clinical question of how to incorporate liquid biopsy into the clinical workflow. To ensure the feasibility of liquid biopsy in oral cancer diagnosis, clinicians and researchers need to cooperate closely to address these and other issues.

## 6. Conclusions

This review discussed recent advances and limitations of liquid biopsy in oral cancer diagnosis. Further development of liquid biopsy technology for oral cancer diagnosis may provide breakthroughs in guiding drug discovery, development, and therapeutic decision-making for personalized medicine.

## Figures and Tables

**Figure 1 jpm-13-00303-f001:**
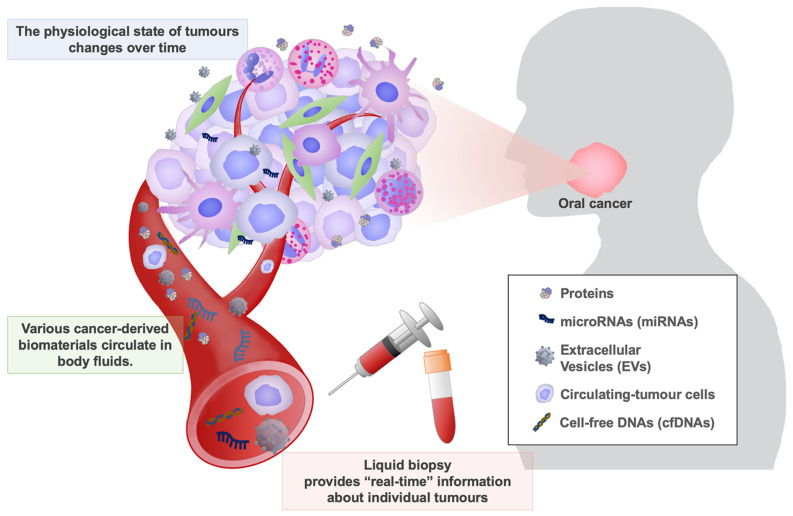
Liquid biopsy for oral cancer diagnosis. Various cancer-derived biomaterials are released into body fluids, such as blood, saliva, and urine. These biomaterials are crucial cues providing “real-time” information about tumours that changes over time. Liquid biopsy can provide such information to support treatment strategies for oral cancer patients.

**Table 1 jpm-13-00303-t001:** Candidate molecules for oral cancer diagnosis presented in this review.

Molecular Target	Target Name	Body Fluid	Purpose	Sensitivity (%)	Specificity (%)	Reference
MicroRNAs	miR-24, miR-20a, miR-122, miR-150, miR-4419a, and miR-5100	Blood (serum)	Cancer detection	55.0	92.5	[20]
	miR-106-5p, miR-423-5p, and miR-193b-3p	Saliva	Cancer detection	85.4	85.1	[21]
Extracellular vesicles (EVs)	miR-210 in EVs	Blood (plasma)	Cancer detection & prognostic marker	92.3	86.6	[22]
	miR-24-3p in EVs	Saliva	Cancer detection	64.4	80.0	[23]
	miR-512-3p and miR-412-3p in EVs	Saliva	Cancer detection	- ^†^	- ^†^	[24]
	Alix in EVs	Blood (serum)	Cancer detection	34.5	100.0	[25]
	Alix in EVs	Saliva	Cancer detection	34.8	100.0	[25]
	miR-503-3p in EVs	Blood (serum)	Prognostic marker	- ^†^	- ^†^	[26]
Cell-free DNAs	Human Arthobacter luteus (ALU) retrotransposon	Saliva	Cancer detection	82.1	70.2	[27]
	beta-2-microglobulin	Saliva	Cancer detection	71.4	81.4	[28]
	Mitochondrial gene	Saliva	Cancer detection	74.2	82.9	[28]
Proteins	CA19-9	Blood (serum)	Cancer detection	- ^†^	- ^†^	[29]
	CEA	Blood (serum)	Cancer detection	66.27	61.21	[29]
	SCC-Ag	Blood (serum)	Cancer detection	73.37	68.10	[29]
	IAP	Blood (serum)	Cancer detection	- ^†^	- ^†^	[30]
	Cyfra	Blood (serum)	Cancer detection	60.36	81.03	[29]

^†^ No corresponding values.

## Data Availability

Not applicable.

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
