# Peer review of "Liquid Biopsy for Oral Cancer Diagnosis: Recent Advances and Challenges"

_jpm, 2023, doi:10.3390/jpm13020303_

Round 1
Reviewer 1 Report
Liquid biopsy for oral cancer diagnosis: recent advances and challenges.
Y. Naito, K. Honda
The authors give a nice review to an important topic in (oral) cancer diagnosis.
The manuscript is well written with nicely prepared figures.
However, I would like to give some comments/ suggestions:
Abstracts: Perhaps it is better to emphasize the importance of liquid biopsy as diagnostic tool by changing the first sentence of the “abstract” from “”Liquid biopsy” is a test used....” into ““Liquid biopsy” is an efficient diagnostic tool used...”.
Introduction: The statements given by the authors in the introduction need to be quoted by references because they are very likely not originally based upon their own observations.
Suggestion: add a new section at the end regarding limitations/problems using analysis of expression levels as diagnostic method. The differences of expression levels do not always correlate with the presence/absence of cancer due to other factors like inflammation. This is mentioned already by the authors. However, it could be explained that this approach is important anyways since it could be used to detect metastasis after tumor excision. Additionally, it could be added that there might be biomarkers which give a yes/no answer since they are expressed only in cancer tissues. These markers are much more interesting because they can give unambiguous diagnostic answers.
Author Response
Response to the comments from Reviewer #1
Q1. Abstracts: Perhaps it is better to emphasize the importance of liquid biopsy as diagnostic tool by changing the first sentence of the “abstract” from “”Liquid biopsy” is a test used....” into ““Liquid biopsy” is an efficient diagnostic tool used...”.
- Thank you very much for your valuable comments. We have changed these sentences as suggested (page 1, line 10).
Q2. Introduction: The statements given by the authors in the introduction need to be quoted by references because they are very likely not originally based upon their own observations.
- We apologize for not citing appropriate references in the introduction section. We have now added citation to proper references in the introduction section.
Following is the text containing the added citations for the statements in the introduction section:
“Recent technological advances in multi-omics platforms have improved our understanding of tumour heterogeneity [1-5]. The genetic background of cancer cells differs significantly among patients. The physiological state of tumours changes over time and is influenced by various factors, including metabolic and homeostatic mechanisms, surrounding microenvironment factors, and drug selection pressures [6]. Such intra-tumoral and inter-patient heterogeneity contributes to treatment failure in patients with cancer [7,8]. Therefore, capturing accurate tumour information in each individual is essential for early detection, determining treatment protocols, predicting disease progression and predicting treatment efficacy.
"Liquid biopsy" is a test for analysing biomaterials (proteins, microRNAs (miRNAs), circulating-tumour cells (CTCs), etc.) in body fluids such as blood, saliva, breast milk, and urine (Figure 1) [9]. Cancer cells and stromal cells within the tumour tissues can secrete various molecules in body fluids [9,10]. Cancer cells detach from tumour tissue during the process of metastasis and circulate in the bloodstream [11]. A biomarker is a factor that is an indicator for normal biological processes, pathogenic processes, or pharmacologic responses to a therapeutic intervention [12]. Biomaterials in body fluids can be considered biomarkers if they reflect such physiological and pathological states of tumours. Liquid biopsy detects these biomarkers, which can include proteins, metabolites, glycans, nucleic acids, and cells in body fluids, and provides "real-time" information about individual tumours (Figure 1). Therefore, liquid biopsy has been studied widely over the past two decades as an attractive diagnostic tool [9,13]. Liquid biopsy is non-invasive and more easily repeatable than conventional histological analysis. So far, many molecular candidates have been investigated for liquid biopsy in cancer diagnosis, such as proteome, microRNAome (miRNAome), glycome, extracellular vesicles (EVs), cell-free DNAs (cfDNAs), and circulating tumour cells (CTCs)[9,10,14].
Head and neck cancer is the seventh most commonly diagnosed solid malignancy worldwide [15,16]. Within head and neck cancer, oral cancer (OC) arises from the oral cavity, including the buccal mucosa, anterior tongue, floor of the mouth, hard palate, upper and lower gingiva, and retromolar trigone. Histologically, approximately 90% of oral cancer is squamous cell carcinoma (SCC) from the mucosal epithelium [17]. Despite improvements in therapeutic strategies, the 5-year survival rate of patients with oral squamous cell carcinoma (OSCC) in later stages (Stages III and IV) remains poor [18,19]. A quick and non-invasive approach, such as liquid biopsy, would be useful for predicting oral cancer metastasis and monitoring treatment efficacy. How could liquid biopsy assist in the diagnosis of oral cancer? This review will focus on recent advances in cancer biomarker research and discuss novel biomarker candidates for liquid biopsy in oral cancer.”
These statements are included on page 1, line 25 through line 45, and page 2, line 67 through line 82.
The references cited in the introduction section are listed below.
References:
- Hudson, T.J.; Anderson, W.; Artez, A.; Barker, A.D.; Bell, C.; Bernabé, R.R.; Bhan, M.K.; Calvo, F.; Eerola, I.; Gerhard, D.S.; et al. International network of cancer genome projects. Nature 2010, 464, 993-998, doi:10.1038/nature08987.
- Crosetto, N.; Bienko, M.; van Oudenaarden, A. Spatially resolved transcriptomics and beyond. Nat Rev Genet 2015, 16, 57-66, doi:10.1038/nrg3832.
- Giesen, C.; Wang, H.A.; Schapiro, D.; Zivanovic, N.; Jacobs, A.; Hattendorf, B.; Schüffler, P.J.; Grolimund, D.; Buhmann, J.M.; Brandt, S.; et al. Highly multiplexed imaging of tumor tissues with subcellular resolution by mass cytometry. Nat Methods 2014, 11, 417-422, doi:10.1038/nmeth.2869.
- Lambrechts, D.; Wauters, E.; Boeckx, B.; Aibar, S.; Nittner, D.; Burton, O.; Bassez, A.; Decaluwé, H.; Pircher, A.; Van den Eynde, K.; et al. Phenotype molding of stromal cells in the lung tumor microenvironment. Nat Med 2018, 24, 1277-1289, doi:10.1038/s41591-018-0096-5.
- Pan, D.; Jia, D. Application of Single-Cell Multi-Omics in Dissecting Cancer Cell Plasticity and Tumor Heterogeneity. Front Mol Biosci 2021, 8, 757024, doi:10.3389/fmolb.2021.757024.
- Hanahan, D.; Weinberg, R.A. Hallmarks of cancer: the next generation. Cell 2011, 144, 646-674, doi:10.1016/j.cell.2011.02.013.
- McGranahan, N.; Swanton, C. Clonal Heterogeneity and Tumor Evolution: Past, Present, and the Future. Cell 2017, 168, 613-628, doi:10.1016/j.cell.2017.01.018.
- Garattini, S.; Fuso Nerini, I.; D'Incalci, M. Not only tumor but also therapy heterogeneity. Ann Oncol 2018, 29, 13-19, doi:10.1093/annonc/mdx646.
- Ignatiadis, M.; Sledge, G.W.; Jeffrey, S.S. Liquid biopsy enters the clinic - implementation issues and future challenges. Nat Rev Clin Oncol 2021, 18, 297-312, doi:10.1038/s41571-020-00457-x.
- Xu, R.; Rai, A.; Chen, M.; Suwakulsiri, W.; Greening, D.W.; Simpson, R.J. Extracellular vesicles in cancer - implications for future improvements in cancer care. Nat Rev Clin Oncol 2018, 15, 617-638, doi:10.1038/s41571-018-0036-9.
- Aoki, M.; Shoji, H.; Kashiro, A.; Takeuchi, K.; Shimizu, Y.; Honda, K. Prospects for Comprehensive Analyses of Circulating Tumor Cells in Tumor Biology. Cancers (Basel) 2020, 12, doi:10.3390/cancers12051135.
- Biomarkers and surrogate endpoints: preferred definitions and conceptual framework. Clin Pharmacol Ther 2001, 69, 89-95, doi:10.1067/mcp.2001.113989.
- Wongsurakiat, P.; Wongbunnate, S.; Dejsomritrutai, W.; Charoenratanakul, S.; Tscheikuna, J.; Youngchaiyud, P.; Pushpakom, R.; Maranetra, N.; Nana, A.; Chierakul, N.; et al. Diagnostic value of bronchoalveolar lavage and postbronchoscopic sputum cytology in peripheral lung cancer. Respirology 1998, 3, 131-137, doi:10.1111/j.1440-1843.1998.tb00111.x.
- Hu, T.; Wolfram, J.; Srivastava, S. Extracellular Vesicles in Cancer Detection: Hopes and Hypes. Trends Cancer 2021, 7, 122-133, doi:10.1016/j.trecan.2020.09.003.
- Chow, L.Q.M. Head and Neck Cancer. N Engl J Med 2020, 382, 60-72, doi:10.1056/NEJMra1715715.
- Mody, M.D.; Rocco, J.W.; Yom, S.S.; Haddad, R.I.; Saba, N.F. Head and neck cancer. Lancet 2021, 398, 2289-2299, doi:10.1016/s0140-6736(21)01550-6.
- Bagan, J.; Sarrion, G.; Jimenez, Y. Oral cancer: clinical features. Oral Oncol 2010, 46, 414-417, doi:10.1016/j.oraloncology.2010.03.009.
- Tiwana, M.S.; Wu, J.; Hay, J.; Wong, F.; Cheung, W.; Olson, R.A. 25 year survival outcomes for squamous cell carcinomas of the head and neck: population-based outcomes from a Canadian province. Oral Oncol 2014, 50, 651-656, doi:10.1016/j.oraloncology.2014.03.009.
- Warnakulasuriya, S. Global epidemiology of oral and oropharyngeal cancer. Oral Oncol 2009, 45, 309-316, doi:10.1016/j.oraloncology.2008.06.002.
Q3. Suggestion: add a new section at the end regarding limitations/problems using analysis of expression levels as diagnostic method. The differences of expression levels do not always correlate with the presence/absence of cancer due to other factors like inflammation. This is mentioned already by the authors. However, it could be explained that this approach is important anyways since it could be used to detect metastasis after tumor excision. Additionally, it could be added that there might be biomarkers which give a yes/no answer since they are expressed only in cancer tissues. These markers are much more interesting because they can give unambiguous diagnostic answers.
- As suggested, we have added a new section at the end of the manuscript to explain the limitations and challenges of the liquid biopsy for oral cancer. The new section is as follows:
“5. Limitations and challenges of novel biomarkers for oral cancer.
As discussed above, increased evidence has shown that liquid biopsy can be a powerful tool for oral cancer diagnosis. For other cancer diagnoses, such as pancreatic cancer and colorectal cancer, several clinical trials have been conducted (Cologuard [20], Galleri test [21], Guardant Health [22]). However, unfortunately, no biomarker is approved by the US Food and Drug Administration (FDA) for oral cancer diagnosis. Tumour lesions can be more readily assessed visually in oral cancer than in other cancer types due to easy access to the oral cavity. Therefore, a liquid biopsy might be less advantageous for detecting tumour lesions and predicting treatment efficacy than other cancers. In the clinical setting, a screening program for oral cancer is not recommended because it is considered that the frequency of oral cancer is very rare. Incidental early detection of oral cancer can occur during dental treatment and oral healthcare. Recently, contactless cancer screening programs have become of interest in the asymptomatic population because of the expansion of novel infectious diseases. If a multi-cancer screening method for not only oral cancer, but also other cancers could be carried out using liquid biopsies during a single clinical examination, it would have a positive impact on precision cancer prevention in a contactless environment. Thus, the establishment of multi-cancer screening methods using liquid biopsies is very important for cancer prevention. Moreover, there is potential for biomarkers to be used for the detection of late metastatic cervical lymph nodes and distant metastases that cannot be seen macroscopically. There are also technical issues, such as regarding how to increase the robustness of the analysis results and the clinical question of how to incorporate liquid biopsy into the clinical workflow. To ensure the feasibility of liquid biopsy in oral cancer diagnosis, clinicians and researchers need to cooperate closely to address these and other issues.”
These statements are included on page 8, line 373 through line 398.
The references cited in this section are listed below.
References:
- Imperiale, T.F.; Ransohoff, D.F.; Itzkowitz, S.H.; Levin, T.R.; Lavin, P.; Lidgard, G.P.; Ahlquist, D.A.; Berger, B.M. Multitarget stool DNA testing for colorectal-cancer screening. N Engl J Med 2014, 370, 1287-1297, doi:10.1056/NEJMoa1311194.
- Neal, R.D.; Johnson, P.; Clarke, C.A.; Hamilton, S.A.; Zhang, N.; Kumar, H.; Swanton, C.; Sasieni, P. Cell-Free DNA-Based Multi-Cancer Early Detection Test in an Asymptomatic Screening Population (NHS-Galleri): Design of a Pragmatic, Prospective Randomised Controlled Trial. Cancers (Basel) 2022, 14, doi:10.3390/cancers14194818.
- Liu, S.; Wang, J. Current and Future Perspectives of Cell-Free DNA in Liquid Biopsy. Curr Issues Mol Biol 2022, 44, 2695-2709, doi:10.3390/cimb44060184.
Reviewer 2 Report
The manuscript gives an insight into the possibilities of liquid biopsy for oral cancer, highlighting the application of an technology that is becoming more and more available to oral cancer.
The manuscript was highly interessting to read and provides a good overview of the current applications and possibilities of liquid biopsy in oral cancer.
Here are the points I identified that require modification:
-) Methods of reference selection and limiations of the study are missing.
-) throughout the manuscript the term "biomaterials" is used instead of "biomarker", this is highly missleading and has to be corrected
-) The abstract is the shortened version of the introduction's second paragraph. I would appreciate reading an abstract covering the entire scope of the paper
-) line 37/38: "nuclear acids" are presumably "nucleic acids"
-) line 49-51: Reference missing
-) 2.1 Conventional cancer markers: The references used for the last paragraph, [10-13] are from 1976-1995. Here a more recent publication is required to ensure that statements are not outdated.
-) 2.2: An explanation of miRNAs is missing
Author Response
Response to the comments from Reviewer #2
Q1. Methods of reference selection and limiations of the study are missing.
- We apologize for not describing the methods for reference selection. We selected the latest manuscripts published within the last two years or with biological evidence in this review. We described the methods for reference selection in the main text as follows.
" In this review, we will additionally focus on the latest findings published within the last two years and those with biological evidence."
These statements are included on page 3, line 104 through line 106.
The limitations of the study are described at the end of this manuscript in the " Limitations and challenges of novel biomarkers for oral cancer " section, as follows:
" 5. Limitations and challenges of novel biomarkers for oral cancer.
As discussed above, increased evidence has shown that liquid biopsy can be a powerful tool for oral cancer diagnosis. For other cancer diagnoses, such as pancreatic cancer and colorectal cancer, several clinical trials have been conducted (Cologuard [20], Galleri test [21], Guardant Health [22]). However, unfortunately, no biomarker is approved by the US Food and Drug Administration (FDA) for oral cancer diagnosis. Tumour lesions can be more readily assessed visually in oral cancer than in other cancer types due to easy access to the oral cavity. Therefore, a liquid biopsy might be less advantageous for detecting tumour lesions and predicting treatment efficacy than other cancers. In the clinical setting, a screening program for oral cancer is not recommended because it is considered that the frequency of oral cancer is very rare. Incidental early detection of oral cancer can occur during dental treatment and oral healthcare. Recently, contactless cancer screening programs have become of interest in the asymptomatic population because of the expansion of novel infectious diseases. If a multi-cancer screening method for not only oral cancer, but also other cancers could be carried out using liquid biopsies during a single clinical examination, it would have a positive impact on precision cancer prevention in a contactless environment. Thus, the establishment of multi-cancer screening methods using liquid biopsies is very important for cancer prevention. Moreover, there is potential for biomarkers to be used for the detection of late metastatic cervical lymph nodes and distant metastases that cannot be seen macroscopically. There are also technical issues, such as regarding how to increase the robustness of the analysis results and the clinical question of how to incorporate liquid biopsy into the clinical workflow. To ensure the feasibility of liquid biopsy in oral cancer diagnosis, clinicians and researchers need to cooperate closely to address these and other issues.”
These statements are included on page 8, line 373 through line 398.
The references cited in this section are listed below.
References:
- Imperiale, T.F.; Ransohoff, D.F.; Itzkowitz, S.H.; Levin, T.R.; Lavin, P.; Lidgard, G.P.; Ahlquist, D.A.; Berger, B.M. Multitarget stool DNA testing for colorectal-cancer screening. N Engl J Med 2014, 370, 1287-1297, doi:10.1056/NEJMoa1311194.
- Neal, R.D.; Johnson, P.; Clarke, C.A.; Hamilton, S.A.; Zhang, N.; Kumar, H.; Swanton, C.; Sasieni, P. Cell-Free DNA-Based Multi-Cancer Early Detection Test in an Asymptomatic Screening Population (NHS-Galleri): Design of a Pragmatic, Prospective Randomised Controlled Trial. Cancers (Basel) 2022, 14, doi:10.3390/cancers14194818.
- Liu, S.; Wang, J. Current and Future Perspectives of Cell-Free DNA in Liquid Biopsy. Curr Issues Mol Biol 2022, 44, 2695-2709, doi:10.3390/cimb44060184.
Q2. throughout the manuscript the term "biomaterials" is used instead of "biomarker", this is highly missleading and has to be corrected
- In this manuscript, we used the term "biomaterials" to refer to the components within various body fluids. In contrast, the term "biomarker" means the factors whose expressions would fluctuate according to the physiological or pathological states of patients or tumours. Indeed, a biomarker is defined as "a characteristic that is objectively measured and evaluated as indicator of normal biological processes, pathogenic processes, or pharmacologic responses to a therapeutic intervention.", according to the Biomarkers Definitions Working Group in the National Institute of Health (NIH) [12]. We need to use these terms appropriately. Therefore, we have added the following sentences:
“ "Liquid biopsy" is a test for analysing biomaterials (proteins, microRNAs (miRNAs), circulating-tumour cells (CTCs), etc.) in body fluids such as blood, saliva, breast milk, and urine (Figure 1) [9]. Cancer cells and stromal cells within the tumour tissues can secrete various molecules in body fluids [9,10]. Cancer cells detach from tumour tissue during the process of metastasis and circulate in the bloodstream [11]. A biomarker is a factor that is an indicator for normal biological processes, pathogenic processes, or pharmacologic responses to a therapeutic intervention [12]. Biomaterials in body fluids can be considered biomarkers if they reflect such physiological and pathological states of tumours. Liquid biopsy detects these biomarkers, which can include proteins, metabolites, glycans, nucleic acids, and cells in body fluids, and provides "real-time" information about individual tumours (Figure 1). Therefore, liquid biopsy has been studied widely over the past two decades as an attractive diagnostic tool [9,13]. Liquid biopsy is non-invasive and more easily repeatable than conventional histological analysis. So far, many molecular candidates have been investigated for liquid biopsy in cancer diagnosis, such as proteome, microRNAome (miRNAome), glycome, extracellular vesicles (EVs), cell-free DNAs (cfDNAs), and circulating tumour cells (CTCs)[9,10,14].”
These statements are included on page 1, line 34 through line 45, and page 2, line 110 through line 113.
The references cited in these sentences are listed below.
References:
- Ignatiadis, M.; Sledge, G.W.; Jeffrey, S.S. Liquid biopsy enters the clinic - implementation issues and future challenges. Nat Rev Clin Oncol 2021, 18, 297-312, doi:10.1038/s41571-020-00457-x.
- Xu, R.; Rai, A.; Chen, M.; Suwakulsiri, W.; Greening, D.W.; Simpson, R.J. Extracellular vesicles in cancer - implications for future improvements in cancer care. Nat Rev Clin Oncol 2018, 15, 617-638, doi:10.1038/s41571-018-0036-9.
- Aoki, M.; Shoji, H.; Kashiro, A.; Takeuchi, K.; Shimizu, Y.; Honda, K. Prospects for Comprehensive Analyses of Circulating Tumor Cells in Tumor Biology. Cancers (Basel) 2020, 12, doi:10.3390/cancers12051135.
- Biomarkers and surrogate endpoints: preferred definitions and conceptual framework. Clin Pharmacol Ther 2001, 69, 89-95, doi:10.1067/mcp.2001.113989.
- Wongsurakiat, P.; Wongbunnate, S.; Dejsomritrutai, W.; Charoenratanakul, S.; Tscheikuna, J.; Youngchaiyud, P.; Pushpakom, R.; Maranetra, N.; Nana, A.; Chierakul, N.; et al. Diagnostic value of bronchoalveolar lavage and postbronchoscopic sputum cytology in peripheral lung cancer. Respirology 1998, 3, 131-137, doi:10.1111/j.1440-1843.1998.tb00111.x.
- Hu, T.; Wolfram, J.; Srivastava, S. Extracellular Vesicles in Cancer Detection: Hopes and Hypes. Trends Cancer 2021, 7, 122-133, doi:10.1016/j.trecan.2020.09.003.
Q3. The abstract is the shortened version of the introduction's second paragraph. I would appreciate reading an abstract covering the entire scope of the paper
- We have revised the abstract as follows.
""Liquid biopsy" is an efficient diagnostic tool used to analyse biomaterials in human body fluids such as blood, saliva, breast milk, and urine. Various biomaterials derived from a tumour and its microenvironment are released into such body fluids and contain important information for cancer diagnosis. Biomaterial detection can provide "real-time" information about individual tumours, is non-invasive, and is more repeatable than conventional histological analysis. Therefore, over the past two decades, liquid biopsy has been considered an attractive diagnostic tool for malignant tumours. Although biomarkers for oral cancer have not yet been adopted in clinical practice, many molecular candidates have been investigated for liquid biopsies in oral cancer diagnosis, such as the proteome, metabolome, microRNAome, extracellular vesicles, cell-free DNAs, and circulating tumour cells. This review will present recent advances and challenges in liquid biopsy for oral cancer diagnosis."(page 1, line 10 through line 20).
Q4. line 37/38: "nuclear acids" are presumably "nucleic acids"
- We have corrected this typographical error on page 1, line 43.
Q5. line 49-51: Reference missing
- We have added citation to the proper references in the introduction section (page 1, line 75 through line 77).
The references we cited for the statements in the introduction section follow below.
References:
- Tiwana, M.S.; Wu, J.; Hay, J.; Wong, F.; Cheung, W.; Olson, R.A. 25 year survival outcomes for squamous cell carcinomas of the head and neck: population-based outcomes from a Canadian province. Oral Oncol 2014, 50, 651-656, doi:10.1016/j.oraloncology.2014.03.009.
- Warnakulasuriya, S. Global epidemiology of oral and oropharyngeal cancer. Oral Oncol 2009, 45, 309-316, doi:10.1016/j.oraloncology.2008.06.002.
Q6. 2.1 Conventional cancer markers: The references used for the last paragraph, [10-13] are from 1976-1995. Here a more recent publication is required to ensure that statements are not outdated.
- Although these publications we cited are not the latest findings, there are few or no papers regarding these conventional cancer markers in oral cancer published in the past two decades. We have revised these sentences to ensure that the information was updated using the latest publications, as follows.
" However, since its accuracy in diagnosis ranges from 64% to 71% [23], these markers are not applicable to all patients with oral cancer." (page 4, line 152 through line 154).
The references we cited for these statements follow.
References:
- Yuan, C.; Yang, K.; Tang, H.; Chen, D. Diagnostic values of serum tumor markers Cyfra21-1, SCCAg, ferritin, CEA, CA19-9, and AFP in oral/oropharyngeal squamous cell carcinoma. Onco Targets Ther 2016, 9, 3381-3386, doi:10.2147/ott.S105672.
Q7. 2.2: An explanation of miRNAs is missing
- We have added an explanation of miRNAs (page 4, line 157).